# Predictors of Free Sugars Intake Trajectories across Early Childhood—Results from the SMILE Birth Cohort Study

**DOI:** 10.3390/ijerph21020174

**Published:** 2024-02-02

**Authors:** Lucinda K. Bell, Huy V. Nguyen, Diep H. Ha, Gemma Devenish-Coleman, Rebecca K. Golley, Loc G. Do, Jane A. Scott

**Affiliations:** 1Caring Futures Institute, College of Nursing and Health Sciences, Flinders University, Adelaide, SA 5042, Australia; rebecca.golley@flinders.edu.au; 2Health Innovation and Transformation Centre, Federation University Australia, Mt. Helen, VIC 3353, Australia; h.vannguyen@federation.edu.au; 3Department of Population and Quantitative Health Sciences, The University of Massachusetts Medical School, Worcester, MA 01655, USA; 4School of Dentistry, Faculty of Health and Behavioural Sciences, The University of Queensland, Brisbane, QLD 4102, Australia; d.ha@uq.edu.au (D.H.H.); l.do@uq.edu.au (L.G.D.); 5School of Population Health, Curtin University, Perth, WA 6907, Australia; gemma.devenish+coleman@gmail.com

**Keywords:** free sugars intake, preschool children, social determinants of health, trajectories

## Abstract

Foods and beverages high in free sugars can displace healthier choices and increase the risk of weight gain, dental caries, and noncommunicable diseases. Little is known about the intake of free sugars across early childhood. This study aimed to examine the longitudinal intake from 1 to 5 years of free sugars and identify the independent maternal and child-related predictors of intake in a cohort of Australian children participating in the Study of Mothers’ and Infants’ Life Events Affecting Oral Health (SMILE). Free sugars intake (FSI) was previously estimated at 1, 2, and 5 years of age, and three distinct FSI trajectories were determined using group-based trajectory modelling analysis. This study utilized multinomial logistic regression to identify the maternal and child-related predictors of the trajectories. The risk of following the ‘high and increasing’ trajectory of FSI compared to the ‘low and fast increasing’ trajectory was inversely associated with socio-economic disadvantage (aRRR 0.83; 95% CI 0.75–0.92; *p* < 0.001), lower for females (aRRR 0.56; 95% CI 0.32–0.98; *p* = 0.042), and higher in children with two or more older siblings at birth (aRRR 2.32; 95% CI 0.99–5.42; *p* = 0.052). Differences in trajectories of FSI were evident from an early age and a high trajectory of FSI was associated primarily with socio-economic disadvantage, providing another example of diet quality following a social gradient.

## 1. Introduction

Early childhood is a period of rapid growth and development. Optimal nutrition is paramount during the first 2 years of a child’s life to support their growth, health, and development [1]. There is concern that the transition during this critical period from a milk-based diet in infancy to a family diet high in free sugars may develop a preference for sweet foods which can track into later childhood and adulthood [2]. While the evidence that frequent and sustained exposure to sweetness in early childhood is associated with a lifelong preference for sweet(er) foods is equivocal [3], foods that are rich in free sugars such as confectionary, cakes, and biscuits, along with sugar-sweetened beverages, can nevertheless displace more nutritionally adequate foods in a child’s diet [4]. Subsequently, this can lead to an unhealthy diet and excessive intakes of free sugars, which has been associated with increased risk of weight gain and noncommunicable diseases [5,6,7]. In addition, excessive and frequent consumption of free sugars is a primary risk factor for dental caries [8]. In light of the health risks associated with the consumption of free sugars, the World Health Organization (WHO) strongly recommends reducing the intake of free sugars to less than 10% of total energy intake in both adults and children [9].

Secondary analysis of the most recent National Nutrition and Physical Activity Survey (NNPAS) conducted in 2011 to 2012 revealed that 68.8% of Australian children aged 2 to 3 years and 80.1% of children aged 4 to 8 years exceeded the WHO recommendation for free sugars intake (FSI) [10]. For children aged 2 to 3 years and 4 to 8 years, the mean daily FSI was estimated to be 42.8 g and 59.7 g, with 12.3% and 13.7% of energy coming from FSI, respectively [10]. However, as the 2011–2012 NNPAS did not include children under the age of 2 years, little is known about intake of free sugars in this younger age group.

Previous cross-sectional analyses of data collected in the Study of Mothers’ and Infants’ Life Events Affecting Oral Health (SMILE) birth cohort reported an usual intake of 8.8 g/day and 22.5 g/day in children aged 1 and 2 years, respectively. At 1 year of age, only 2.4% of participants exceeded the WHO recommendation that <10% of energy should come from free sugars, but by 2 years, this proportion had increased markedly to 38% [11,12]. Subsequently to these analyses, a third wave of dietary data was collected in the SMILE study when children were aged 5 years, which enabled the use of group-based trajectory modelling (GBTM) to characterize trajectories of FSI from 1 to 5 years of age [13].

The GBTM method identifies clusters of individuals who follow similar trajectories of health behaviors over time [14,15]. While cross-sectional analyses can provide some indication of how dietary quality or the intake of different foods and nutrients evolves at a population level, longitudinal trajectory analysis can identify whether individual children exhibit varying trajectories of intake. Trajectory analysis of dietary data collected in longitudinal studies is becoming increasingly popular and being used to predict a variety of outcomes in childhood and adolescence [16,17,18,19]. GBTM has been used to characterize trajectories of sugar intake in Brazilian preschoolers from 4 to 48 months [20] and children from 4 to 18 years [21]. To our knowledge, however, the SMILE study is the first to report on longitudinal trajectories of FSI in the critical developmental period of early childhood in Australian children [13].

We have previously reported on the predictors of FSI in the SMILE cohort at ages one and two years in cross-sectional analyses [11,12]. This study aimed to identify the independent maternal and child-related predictors of the longitudinal trajectories of FSI: (1) ‘low and fast increasing’, (2) ‘moderate and increasing’, and (3) ‘high and increasing’ [13].

## 2. Materials and Methods

### 2.1. Data Source

Data for this analysis were collected in the Study of Mothers’ and Infants’ Life Events Affecting Oral Health (SMILE), an ongoing population-based birth cohort study in Adelaide, South Australia. SMILE has collected data from birth on a variety of factors (such as diet, socioeconomic status, and health behaviors) along with the dental health of the children at different ages across early childhood. The study protocol and recruitment procedures [22] have been described in detail elsewhere. Briefly, between July 2013 and August 2014, 2181 mother/infant dyads were recruited from three major maternity hospitals. Participants were followed up with questionnaires when the child was 3 and 6 months of age, and 1, 2, and 5 years of age. Oral epidemiological examinations and anthropometric assessments were conducted at age 2 and 5 years. Data used in this analysis were collected at birth and from three waves when children were aged 1 year (2014–2015), 2 years (2015–2016), and 5 years (2018–2019).

### 2.2. Key Measures

#### 2.2.1. Outcome Variable—Trajectories of Children’s Free Sugars Intake

Child’s usual FSI (grams per day) was estimated at ages 1, 2, and 5 years [13]. At age 1 year, 3 days of non-consecutive dietary intake data were collected for each child using a 24 h recall and 2-day food record, entered into FoodWorks Version 8 (Xyris Software, 2012–2017), and FSI was estimated using the Australian food composition database, AUSNUT 2011–13 [12,23]. At ages 2 and 5 years, age-specific 98- and 99-item Food Frequency Questionnaires (FFQ) were developed to collect detailed data on foods and beverages containing free sugars [24]. Frequency and quantity response options appropriate to these age groups were developed for each item. Seven frequency response options were used for all items, ranging from ‘never or rarely’ to ‘3 or more times per day’. Quantity response options were customized to each item, comprising household measures (teaspoon, tablespoon, cup, etc.) or typical portion sizes (piece, tub, pouch, etc.). Finally, the SMILE-FFQ was analyzed using a customized database which linked scoring algorithms for all possible frequency responses to grams of free sugars, derived from representative foods in the AUSNUT 2011–13 food composition database [11,23]. The validity of, and consistency between, the SMILE-FFQ and three nonconsecutive 24 h recalls have been demonstrated previously in an external cohort [24].

We have previously described the method of GBTM which was applied to child FSI intake data at 1, 2, and 5 years of age to generate trajectories of FSI intake [13]. Three methods for handling missing data, which is unavoidable in longitudinal studies [25], were applied to the data: firstly, the pairwise method, which employs all cases with at least one data point (*n* = 1386); then the modified pairwise method, employing all cases with at least two data points (*n* = 1019); and finally the listwise method, employing only cases with all three data points (*n* = 546). Each method generated three similar trajectories, and trajectory membership was comparable across the three approaches [13].

Given the consistency of the findings across the three methods, the FSI trajectories derived from the pairwise method (Figure 1) and corresponding sample were used in this analysis to optimize the sample size and statistical power. Some 16.6% of children were characterized as having a ‘high and increasing’ trajectory of FSI, characterized by having the highest level of intake at 1 year of age, with this continuing to increase over the follow-up period. A further 15.1% of children had a low early intake that increased markedly during the second year of life. This trajectory was categorized as ‘low and fast increasing’. The remaining majority (68.3%) had a moderate level of intake initially that continued to increase, i.e., the ‘moderate and increasing’ trajectory. An ordinal variable representing trajectory group membership was used here as the outcome measure.

#### 2.2.2. Explanatory Variables

The sociodemographic variables used in these analyses were those identified in earlier analyses of the SMILE cohort data [11,12], and other studies [16,26] as being independent determinants of dietary intake in early childhood. Maternal and household characteristics included the mother’s age at time of delivery (years); the mother’s educational attainment (High School, Vocational Training, Tertiary); the mother’s country of birth (Australia/New Zealand, UK, India, China, Asia (other), Others); the mother’s pre-pregnancy body mass index (BMI) (<25, 25–29.99, ≥30 kg/m^2^); and household type at birth (single- or two-parent household). Residential postcodes were used to assign an ordinal measure of socio-economic position using the Index of Relative Socio-Economic Advantage and Disadvantage (IRSAD) deciles, where 1 and 10 ranked as most socio-economically disadvantaged and most socio-economically advantaged, respectively [27]. Child-related factors collected at baseline included sex (female, male), birthweight (<2500 g, 2500–4000 g, >4000 g), number of older siblings (0, 1, ≥2), duration of breastfeeding (<17, 17–25, 26–51, ≥52 weeks), and age of introduction of complementary (solid and semisolid) foods (<17, 17–25, ≥26 weeks).

### 2.3. Statistical Analysis

Data were analyzed using IBM SPSS Statistics for Windows, Version 28.0 (IBM Corp: Armonk, NY, USA). The dependent outcome variable in this analysis was trajectory of FSI generated by the pairwise approach. As the outcome variable has more than two categorical levels, a simple multinomial logistic regression analysis was initially conducted to investigate the association of the individual explanatory variables and trajectory of FSI (Appendix A). The reference category for the outcome variable was trajectory 1 (‘low and fast increasing’). Then, a multivariable multinomial logistic regression analysis was conducted to investigate the independent association of the explanatory variables and trajectory of FSI. Explanatory variables with a *p* < 0.10 for the bivariate analysis were entered into the full model. The model fit was assessed by the likelihood ratio test. The exponentiated results of the multinomial logistic regression are reported as adjusted relative risk ratios (aRRRs), and 95% confidence intervals (CIs). Sensitivity analysis was conducted to test the consistency of estimates using the trajectories of FSI generated by the listwise approach to GBTM. A *p*-value less than 0.05 was considered statistically significant.

## 3. Results

Overall, the FSI increased rapidly in the second year of life and thereafter continued to rise but at a slower rate. The three trajectories, while never crossing, tended to converge at two years of age. Based on the pairwise sample (*n* = 1386), the mean FSI at one year of age was 8.8 g versus 32.2 g at two years, and 44.2 g at five years (Table 1). There was an increasing non-linear trend in FSI over time (*p* < 0.001) for the total sample and for each trajectory.

After adjusting for covariates (Table 2), an increasing level of socio-economic advantage at birth (aRRR 0.91; 95% CI 0.85–0.98; *p* = 0.013) and shorter durations of breastfeeding compared with breastfeeding to 52 weeks and beyond were associated with a higher risk of following trajectory 2—‘moderate and increasing’—compared to trajectory 1—‘low and fast increasing’—while mother’s age (aRRR 0.94; 95% CI 0.89–1.00, *p* = 0.047), being female (aRRR 0.55; 95% CI 0.32–0.97; *p* = 0.040), and an increasing level of socio-economic advantage (aRRR 0.84; 95% CI 0.75–0.93; *p* < 0.001) were associated with a lower risk of following trajectory 3—‘high and increasing’. Conversely, children with two or more older siblings (aRRR 2.31; 95% CI 0.98–5.43) were at greater risk of following trajectory 3 compared to children with no older siblings, although this association just failed to reach statistical significance (*p* = 0.055).

Sensitivity analysis using trajectories generated by the listwise approach (*n* = 546) to GBTM (Appendix A) showed results similar to those presented for risk of following trajectory 3 compared to trajectory 1 with regard to mother’s age (aRRR 0.90; 95% CI 0.82–0.98, *p* = 0.011), being female (aRRR 0.45; 95% CI 0.21–0.99; *p* = 0.047), and socio-economic advantage (aRRR 0.83; 95% CI 0.72–0.96; *p* = 0.009) and was significantly associated with having two or more older siblings (aRRR 10.65; 95% CI 2.55–44.38; *p* = 0.001). Risk of following trajectory 2 compared to trajectory 1 was inversely associated with increasing level of socio-economic advantage (aRRR 0.89; 95% CI 0.80–0.98; *p* = 0.18) and children with two or more older siblings were at significantly higher risk (aRRR 4.41; 95% CI 1.28–15.23; *p* = 0.019) compared to children with no older siblings. The association with duration of breastfeeding was not apparent in the sensitivity analysis.

## 4. Discussion

We have previously described the process for identifying three distinct trajectories of FSI in children from 1 to 5 years of age [13]. The majority of children (68.3%) followed the ‘moderate and increasing’ trajectory and a roughly equivalent proportion of children followed the ‘low and fast increasing’ (15.1%) and ‘high and increasing’ (16.6%) trajectories. These trajectories were primarily distinguished by differences in the amount of FSI consumed at 1 year of age. When compared to children in the ‘low and fast increasing’ trajectory group, increasing level of socio-economic advantage was inversely associated with risk of being in either the ‘moderate and increasing’ or ‘high and increasing’ groups. Duration of breastfeeding was inversely associated with the risk of being in the ‘moderate and increasing’ group, while increasing maternal age and being a girl were associated with a lower risk of following the ‘high and increasing’ trajectory. There was a noticeable trend for children with two or more older siblings to be at greater risk of following the ‘high and increasing’ trajectory.

The maternal and child characteristics predictive of trajectories of FSI in this analysis are broadly consistent with those identified in earlier cross-sectional analyses of the SMILE data at 1 and 2 years of age [11,12]. At these age points, higher intakes of free sugars were directly associated with higher levels of socio-economic disadvantage at birth in the 1 and 2 years analyses and with younger maternal age and having two or more older siblings at birth in the 2 years analysis, whereas only at 1 year of age were female children of mothers born in Australia or New Zealand less likely to have higher intakes of free sugars than male children of mothers born in Australia or New Zealand. We did not see in either the primary or sensitivity analyses the associations with maternal level of education or country of birth and FSI reported in the cross-sectional analysis at 2 years of age.

The findings of this study are yet another example of diet quality following a social gradient in Australia, and internationally [28,29]. Compared to trajectory 1, the risk of children following either a ‘moderate increasing’ or ‘high increasing’ trajectory was inversely associated with level of socio-economic advantage. Consistently with our findings, Manohar et al., who used GBTM to characterize early childhood trajectories of discretionary food intake in toddlers from South Western Sydney, reported that socio-economic disadvantage was associated with high trajectories of discretionary foods intake [16]. Similar socio-economic disparities in the diet quality of children have been reported in other Australian studies [26,30,31]. Higher-quality diets are associated with greater affluence, whereas energy-dense diets that are nutrient poor are associated with lower socio-economic position, and this relationship is consistently observed in different groups by age and sex [29]. This reinforces that the determinants of healthy eating are more complex than a lack of knowledge and cooking skills. Discretionary foods and beverages which are high in saturated fat and added sugar or salt are highly palatable, heavily marketed to children, readily available and widely consumed in Australia and other high and middle-income countries [28]. In addition, these foods tend to cost less per calorie, compared to heathier, nutrient-dense foods [32]. If the primary driver of these disparities is a lack of economic resources, then health promotion strategies which focus on recommending high-cost foods are likely to be ineffective.

We found a significant relationship between trajectories of FSI and sex, with males being twice as likely to follow the ‘high increasing’ FSI trajectory compared to females. Similarly, Manohar and colleagues reported that females were 36% less likely to fall into the highest trajectory for discretionary food and beverage intake than males. Our findings, and those of other studies [33,34,35], suggest that at least some of the dietary disparities between females and males that are reported in later childhood and adolescence [36,37], may have their origins in early childhood. These findings, however, are inconsistent [38] and somewhat difficult to explain in this age group when the provision of food is primarily controlled by parents compared to older age groups when children have greater agency over their food choices. Nevertheless, there is evidence that gendered food habits persist among adults [39], that gendered body ideals are internalized from a young age [40], and that parental eating psychopathology can result in gendered feeding behaviors, in particular by mothers towards their daughters [41].

In this study, children with two or more older siblings were twice as likely to follow a ‘high increasing’ FSI trajectory than a ‘low and fast increasing’ FSI trajectory. While this association just failed to reach statistical significance, it was statistically significant in the sensitivity analysis. Similarly, Manohar et al. reported that children were more likely to follow higher trajectories of discretionary foods intake in early childhood if they lived in households with three or more children [16]. Our finding that children with two or more older siblings were more likely to follow high trajectories of FSI than first born children is consistent with other Australian [26] and international [34,42,43] studies which have reported poorer diet quality and dietary patterns among children with older siblings.

Older children in the family who are exposed to discretionary foods outside of the home and through food advertising may pester their parents for these foods, influencing and potentially increasing the availability of these foods in the household. Additionally, caregivers with larger families may be more time-poor and therefore rely more on convenience foods which are typically energy-dense and nutrient-poor [43]. There have been calls in Australia for Government regulation to specifically protect children from unhealthy food marketing [44] and for improved nutrition labelling of added sugars. These actions, if implemented, may make it easier and less stressful for parents to identify healthier products and make healthier family food choices.

To date, the only other study to characterize and report on trajectories of sugar consumption in early childhood is the 2015 Pelotas Birth Study (PBS) in Brazil, which used GBTM on data collected at 3, 12, 24, and 48 months of age [20]. The PBS identified four distinct trajectories of sugar consumption: ‘sugar consumption always low’, ‘sugar consumption always intermediate’, ‘increasing sugar consumption’ (reflecting consumption that started low and increased significantly as the child got older), and ‘sugar consumption always high’ (reflecting consumption that was consistently high during the first 4 years of life). While we identified a trajectory which started high and remained high, the trajectory which started low at 12 months rose rapidly in the second year of life and, while trajectories never crossed, this trajectory converged on the high and moderate trajectories at 2 years of age.

The dissimilarities in the trajectories reported in the PBS and the SMILE study are most likely explained by the methods used to estimate sugar intake at each age point. In the SMILE study, we employed widely-used and accepted dietary assessment methods [45], which included a customized and validated multi-item FFQ [24] to estimate usual intake of free sugars in grams per day, whereas the PBS recorded consumption of a limited number of foods and beverages containing added sugars at each age point to derive a proxy measure of sugar consumption. The PBS was not designed to quantify usual intake of free sugars per se, which is a notable strength of the SMILE study that, from the outset, was designed to comprehensively measure early feeding practices and free sugars intake at ages 1, 2, and 5 years [22].

The primary limitation of this study is attrition and associated missing data, which affects all longitudinal studies [25]. As predicted in the planning of SMILE, this was most apparent amongst those from a socio-economically disadvantaged background, and attempts were made to minimize this limitation by over-sampling this group at recruitment [22] and in the methods used to generate the FSI trajectories [13]. The SMILE study is the first to use GBTM to identify longitudinal trajectories of FSI in the critical developmental period of early childhood in Australian children, and report on associated maternal and child-related predictors. This is a novel contribution to the literature and a key strength of the present study.

## 5. Conclusions

In this study, differences in trajectories of FSI were evident from an early age, and a high trajectory of FSI was associated primarily with socio-economic disadvantage and number of older siblings. Strategies and policies which target the social determinants of health are likely to be more effective in reducing the intake of free sugars than simply telling individuals to eat healthier foods.

## Figures and Tables

**Figure 1 ijerph-21-00174-f001:**
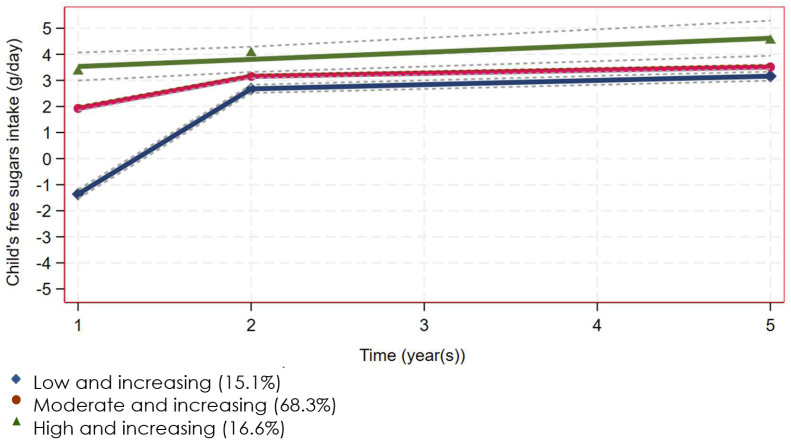
Trajectories of child’s free sugars intake from ages 1 to 5 years (*n* = 1386).

**Table 1 ijerph-21-00174-t001:** Children’s free sugars intake (g/d) at 1, 2, and 5 years of age by trajectory of intake (*n* = 1386).

	Trajectory 1 (*n* = 165)	Trajectory 2 (*n* = 1095)	Trajectory 3 (*n* = 126)	Total (*n* = 1386)
Child age	Mean (SD)	Mean (SD)	Mean (SD)	Mean (SD)
1 year old	0.31 (0.22)	7.91 (6.22)	30.50 (24.82)	8.8 (12.0)
2 years old	19.66 (15.45)	28.40 (23.09)	80.91 (85.19)	32.2 (37.8)
5 years old	18.27 (18.65)	28.92 (31.99)	87.29 (76.65)	44.2 (45.7)
*p* _-Mann-Kendall trend_	<0.001	<0.001	<0.001	<0.001

Trajectory 1: low and fast increasing; trajectory 2: moderate and increasing; trajectory 3: high and increasing.

**Table 2 ijerph-21-00174-t002:** Maternal and child characteristics at birth associated with trajectory of free sugars intake pairwise—adjusted risk (*n* = 1085).

	Trajectory 2 Compared with Trajectory 1			Trajectory 3 Compared with Trajectory 1		
	aRRR	95% CI	*p* Value	aRRR	95% CI	*p* Value
**Maternal characteristics**						
**Mother’s age**	1.01	0.96–1.05	0.806	**0.94**	**0.89–1.00**	**0.047**
**Mother’s highest education level**						
High school (*n* = 191)	1.00	0.54–1.84	0.991	1.51	0.68–3.36	0.315
Vocational training (*n* = 294)	0.78	0.50–1.22	0.278	0.75	0.38–1.48	0.403
Tertiary education (*n* = 600)	REF			REF		
**IRSAD decile**	**0.91**	**0.85–0.98**	**0.013**	**0.84**	**0.75–0.93**	**<0.001**
**Household composition at birth**						
Single-parent household (*n* = 63)	1.69	0.59–4.86	0.328	3.13	0.92–10.66	0.068
Two-parent household (*n* = 1022)	REF			REF		
**Child characteristics**						
**Child sex**						
Male (*n* = 577)	REF			REF		
Female (*n* = 508)	0.85	0.58–1.24	0.392	**0.55**	**0.32–0.97**	**0.040**
**Number of older siblings**						
None (*n* = 531)	REF			REF		
One (*n* = 384)	1.35	0.88–2.08	0.175	1.36	0.72–2.59	0.346
Two or more (*n* = 170)	1.60	0.86–2.98	0.137	2.31	0.98–5.43	0.055
**Duration of breastfeeding (weeks)**						
<17 (*n* = 334)	**1.69**	**1.01–2.84**	**0.048**	1.62	0.78–3.40	0.199
17–25 (*n* = 106)	1.33	0.68–2.60	0.405	1.02	0.37–2.85	0.969
26–51 (*n* = 218)	**1.71**	**1.00–2.91**	**0.049**	1.84	0.84–4.04	0.128
≥52 (*n* = 427)	REF			REF		
**Age of introduction of complementary foods (weeks)**						
<17 (*n* = 275)	1.32	0.61–2.83	0.480	1.55	0.54–4.46	0.420
17–25 (*n* = 707)	0.99	0.53–1.88	0.985	0.71	0.28–1.83	0.481
≥26 (*n* = 103)	REF			REF		

Likelihood ratio test chi-square = 65.908; df = 26, *p* < 0.001. Trajectory 1: low and increasing (*n* = 127, 11.7%); trajectory 2: moderate and increasing (*n* = 863, 79.5%); trajectory 3: high and increasing (*n* = 95, 8.8%). IRSAD: Index of Relative Socio-Economic Advantage and Disadvantage, where decile 1 = most socially disadvantaged, and decile 10 = most socially advantaged.

## Data Availability

The data presented in this study are available upon reasonable request from the corresponding authors. The data are not publicly available due to ethical restrictions.

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
