# Peer review of "Predictors of Free Sugars Intake Trajectories across Early Childhood—Results from the SMILE Birth Cohort Study"

_ijerph, 2024, doi:10.3390/ijerph21020174_

Round 1
Reviewer 1 Report
Comments and Suggestions for Authors
This study used GBTM to identify longitudinal trajectories of FSI in the critical developmental period of early childhood in Australian children, and report on associated maternal and child-related predictors.
I have just some minor comments:
The place of the study (Australia) should be mentioned in the title or abstract.
In abstract:
Please mention the duration that children have been studied in this cohort.
The conclusion has not driven by the results of this study.
The conclusion needs to be rewritten.
Author Response
Thank you for your suggestions.
The place of the study has been identified in the abstract.
The duration over which the free sugars intake of children (i.e. 1 to 5 years) has been added to the abstract.
The conclusion has been rewritten.
Reviewer 2 Report
Comments and Suggestions for Authors
This well-designed longitudinal study contributes to understanding how free sugar intake develops among Australian children between 1-5 y. The primary outcome and confounders are well-researched and properly discussed.
a few comments:
46-48: The updated WHO publication (2023) on carbohydrate intake for adults and children should be mentioned (ISBN 978-92-4-007359-3)
133, fig 1: please describe the units: child-free sugar intake (y-axis)
150: Birth weight is among the most important predictors of toddler and preschool growth/overweight and obesity. Birthweights <2500g and >4000g are not in the normal range. Birthweights between 2500g and 4000g represent the normal range (WHO). Therefore, this explanatory variable should have 3 categories (presentation and analysis) with birthweight between 2500 - 4000g as the reference (supplementary tables 1-3). It does not make sense to have birthweight <2500g as the reference because low birthweight is often associated with postnatal disease and hospitalization.
152 Duration of breastfeeding/introduction of complementary food. Those are very important explanatory variables that have been associated with later obesity in many studies. WHO recommends exclusive breastfeeding until 6 months but scientific nutrition committees disagree (eg. ESPGHAN 4-6m). As an alternative to presenting the variable breastfeeding duration in weeks after birth, 3 categories of the variable could be considered: BF <4mo, 4-6mo, and > 6mo (as the reference). The introduction of complementary feeding could be presented the same way with >6mo as the reference. With this approach, it could be tested if the present global feeding recommendations (WHO) can contribute to lower free sugar intake between 1 and 5 years of age. 43% of Australian infants still receive breast milk at 12 months of age. It would be of interest if the high breastfeeding rate at that age contributes to the very low intake of free sugars at 1 y, in particular in the low and increasing FSI trajectory (0.31g/d at 1y !).
171 the second year of life was analyzed and not the first year
Author Response
46-48: The updated WHO publication (2023) on carbohydrate intake for adults and children should be mentioned (ISBN 978-92-4-007359-3)
Response: We have added cited (reference 7) the 2023 WHO guideline Carbohydrate intake for adults and children in the introduction as suggested, but not in relation to the recommended intake of free sugars as a percentage energy, as the new guideline does not make a specific quantitative recommendation related to free sugars intake.
133, fig 1: please describe the units: child-free sugar intake (y-axis)
Response: the unit of measurement is now included on the y-axis
150: Birth weight is among the most important predictors of toddler and preschool growth/overweight and obesity. Birthweights <2500g and >4000g are not in the normal range. Birthweights between 2500g and 4000g represent the normal range (WHO). Therefore, this explanatory variable should have 3 categories (presentation and analysis) with birthweight between 2500 - 4000g as the reference (supplementary tables 1-3). It does not make sense to have birthweight <2500g as the reference because low birthweight is often associated with postnatal disease and hospitalization.
Response: We have recategorized birth weight into 3 groups as recommended, using 2500-4000g as the reference category and reanalysed the data. Again, there was no significant association in the bivariate analysis between birthweight and FSI trajectories in either the pairwise or listwise (sensitivity) analysis.
152 Duration of breastfeeding/introduction of complementary food. Those are very important explanatory variables that have been associated with later obesity in many studies. WHO recommends exclusive breastfeeding until 6 months but scientific nutrition committees disagree (eg. ESPGHAN 4-6m). As an alternative to presenting the variable breastfeeding duration in weeks after birth, 3 categories of the variable could be considered: BF <4mo, 4-6mo, and > 6mo (as the reference). The introduction of complementary feeding could be presented the same way with >6mo as the reference. With this approach, it could be tested if the present global feeding recommendations (WHO) can contribute to lower free sugar intake between 1 and 5 years of age. 43% of Australian infants still receive breast milk at 12 months of age. It would be of interest if the high breastfeeding rate at that age contributes to the very low intake of free sugars at 1 y, in particular in the low and increasing FSI trajectory (0.31g/d at 1y !).
Response: We have categorised breastfeeding duration and age of introduction of solids. With regards to BF duration, we created 4 categories with ≥ 52 weeks being the reference category as the Australian Infant Feeding Guidelines recommend that infants should be breastfed to 12 months or longer.
The results of the analysis using these variables as categorical rather than continuous variables did not yield markedly different results. Age of introduction of solids remained insignificant and duration of breastfeeding remained a significant predictor of trajectory 2 versus trajectory 1 and was not a significant predictor of trajectory 3 versus trajectory 1.
The significant predictors identified in the original analysis remained unchanged with the exception that maternal age which was near to significance in the original analysis is now a significant predictor of Trajectory 3 versus trajectory 1.
Table 1 and the supplementary tables have been updated with the new effect size estimates.
Line 171, thank you for picking up that rapid increase in FSI occurred in the second year of life.
Reviewer 3 Report
Comments and Suggestions for Authors
The methods and results are very well presented.
Only one small question, low socioeconomic status and families have more than one child have higher consumption of free sugar, can we use food labels to change this situation? or what should we do?
Please discuss more about it.
Author Response
Only one small question, low socioeconomic status and families have more than one child have higher consumption of free sugar, can we use food labels to change this situation? or what should we do?
Response
We offer improved food labelling as one alternative which might make it easier for time poor parents of larger families to recognise foods that are high in free sugars. Current labelling requirements in Australia related to added sugars are inadequate, although they are likely to be tightened in the near future.
Nevertheless, while we call for improved labelling in the discussion as a way to help parents we have not referred to food labelling policies in the conclusion.
We acknowledge that clearer labelling would not address the socio-economic diet gradients.
Round 2
Reviewer 3 Report
Comments and Suggestions for Authors
N/A